# Leather Waste Hydrolysation, Carbonization, and Microbial Treatment for Nitrogen Recovery by Ryegrass Cultivation

**DOI:** 10.3390/ma17235741

**Published:** 2024-11-23

**Authors:** Ksawery Kuligowski, Dawid Skrzypczak, Katarzyna Mikula, Katarzyna Chojnacka, Paulina Bandrów, Robert Tylingo, Szymon Mania, Adrian Woźniak, Adam Cenian

**Affiliations:** 1Department of Physical Aspects of Ecoenergy, The Institute of Fluid-Flow Machinery Polish Academy of Sciences, Fiszera 14 St., 80-231 Gdańsk, Poland; 2Department of Advanced Material Technology, Faculty of Chemistry, Wroclaw University of Science and Technology, M. Smoluchowskiego 25 St., 50-372 Wroclaw, Poland; 3BADER Polska Sp. z o.o., Mostowa 1 St., 59-700 Bolesławiec, Poland; 4Department of Chemistry, Technology and Biotechnology of Food, Faculty of Chemistry, Gdańsk University of Technology, Narutowicza 11/12, 80-233 Gdańsk, Poland; 5Rendben Ltd., Wiczlińska 117 M, 81-578 Gdynia, Poland

**Keywords:** tanned leather waste, hydrolysates, biochar, ryegrass growth, agronomic effectiveness, nitrogen use efficiency

## Abstract

Leather waste contains up to 10% nitrogen (N); thus, combustion or gasification only for the energy recovery would not be rational, if safety standards are met. On the other hand, the chromium (Cr) content exceeding 5% in half of the waste stream (*w*/*w*) is too significant to be applied in agriculture. In this work, four acid hydrolysates from leather waste shavings, both wet-white free of Cr and wet-blue with Cr, were used: two with a mixture of acids and supplemented with Cu, Mn, and Zn, and the other two as semi-products from collagen extraction using hydrochloric acid. Additionally wet-green leather waste shavings, e.g., impregnated with olive extract, were used followed by the two treatments: amendment with a biochar from “wet white” leather waste shavings and amendment with this biochar incubated with the commercial phosphorus stimulating microbial consortia BactoFos. They were applied as organic nitrogen-based fertilizers in a glasshouse experiment, consisting of 4–5 subsequent harvests every 30 days, under spring–autumn conditions in northern Poland. Biochar-amended wet-greens provided the highest nitrogen use efficiencies, exceeding 100% after 4 months of growth (for 20 kg N/ha) and varying from 17% to 37% in particular months. This is backed up by another parameter (relative agronomic effectiveness) that for these materials exceeded 150% for a single month and in total was around 33%. Biochar amendments significantly increased agronomic parameters for wet-greens, and their microbial treatment enhanced them even further. Recycling this type of waste can replace inorganic fertilizers, reducing greenhouse gas emissions and carbon footprint.

## 1. Introduction

Product and processing modifications can contribute to innovation and competitiveness in various industrial sectors, economic growth, and the creation of new workplaces, which pays off for both entrepreneurs and consumers [1]. Due to the aforementioned significant environmental and economic benefits, the European Union is encouraging economic transformation. This is further prompted by the fact that most raw materials are imported into the EU, and recycling waste could alleviate issues related to availability and price [2].

Recycling leather waste has gained importance due to the environmental impact of leather production and disposal. Leather scraps and waste from manufacturing, end-of-life products, or even defective items can be recycled or repurposed in several innovative ways. Among several examples, the following ones deserve special attention since they are still at the test stage: biodegradable leather alternatives [3,4,5,6,7], leather-based biochar and fertilizers [8,9,10], and collagen extraction [11,12,13,14,15,16,17]. Recycling leather waste is complex due to its durability, chemical treatments, and varying quality. However, with growing sustainability awareness, recycling methods are becoming more advanced, paving the way for leather alternatives and upcycled leather products. These methods not only reduce waste but also offer new materials that can decrease the industry’s environmental footprint.

### 1.1. Closing the Loop in Tanning Waste

Changing from a linear to a circular model would reduce costs associated with waste disposal and emissions control and improve the public image of the tanning business due to being more environmentally friendly [1]. Solid tannery wastes can be an energetic secondary feedstock because of their high heating value (17–20 MJ/kg), but high-temperature treatment can lead to the release of nitrogen oxides [18] or the formation of toxic Cr(VI) [19]. Tannery shavings and sludge have potential to be reused to produce biogas, which could meet growing energy needs and simultaneously reduce fossil fuel consumption [20]. Tannery waste is a source of fats, which can be utilized in biofuel production using supercritical ethanol [21]. It is possible to obtain proteases from tannery waste that have applications in medicine, biotechnology, or the food industry. For this purpose, the waste is subjected to biological degradation using *Synergistes* sp., Pseudomonas aeruginosa, and Selenomonas ruminantium [22]. Treating the remains of goat skins with organic acids (such as acetic or propionic acid) can provide type I collagen, which is a valuable substance in pharmacy and cosmetics [23]. Collagen from leather waste can also be a component of hybrid polymer films for medical applications [24], an ingredient in high-strength composites with Al_2_O_3_ [11], and even building materials that retard the bonding of gypsum [25]. The agricultural sector can also be a recipient of processed tannery waste. Tanning waste provide a renewable source of organic nitrogen (protein), which can be converted into short peptides and free amino acids through hydrolysis [26]. Amino acids such as proline, glycine, arginine, lysine, valine, aspartic acid, and glutamic acid can be present in hydrolysed leather trimmings, which promote healthy plant development [27].

Different treatment methods (thermal, biological, and chemical) recover valuable components from tannery residues and generate new and useful products for various industries. However, the ongoing fertilizer crisis (related to the availability of raw materials, the high price of natural gas, and, consequently, the high price of commercial products) makes the exploration of alternative raw materials for the production of agrochemicals seem particularly important [28].

### 1.2. Hydrolysis and/or Carbonization?

Hydrolysates based on leather waste are an intermediate for the preparation of fertilizers. They are obtained as a result of temperature, enzymes, or chemicals. The last type of hydrolysis is most frequently practised because it is inexpensive, easy, and quick [29]. Various mineral acids (H_2_SO_4_, H_3_PO_4_, and HCl), organic acids (acetic and citric), and acid mixtures or alkalis (mainly NaOH or CaOH) are employed as hydrolysing agents [30]. Considering the fertilizer direction of waste valorization, it is most beneficial to apply phosphoric acid (V) or potassium hydroxide for treatment. The achieved hydrolysate is a more valuable product, rich not only in nitrogen (in the form of amino acids and short peptides) but also in the essential macronutrients P and K [26]. Fertilizers based on hydrolysate from wet-blue shavings mixed with poultry bone meal and hyacinth ash [31], or by mixing hydrolysate with ash (from biomass combustion or wastewater treatment plants), have been successfully produced [26]. Fertilizer was also made as compost from tannery sludge mixed with manure and straw [32]. Carbonization of tanned leather waste fractions via pyrolysis has already been studied, but the application of wet-greens as a novel feedstock material, amendment with biochar, followed by microbial treatment constitutes a novel approach.

The valorization of tanning wastes for fertilizer purposes has not yet been widely explored. Residues from tanning leather with chromium(III) salts are mainly reused for the production of agrochemicals [33]. After fertilizer application, chromium can be released into the soil and accumulate there or in the plant, causing a threat to the environment and future food consumers. In vivo usefulness studies (pot tests, greenhouse tests, and field tests) are particularly important in this case. They provide information about the assimilation of nutrients from the fertilizer into the plant and also verify the possibility of transferring hazardous contaminants into the environment. Additionally, the tanning industry is opening up to green technologies and using plant extracts for leather processing. There is a lack of information on their potential use in the agricultural sector.

### 1.3. The Objective of the Study

The objective of this study was to investigate the effects of various treated tannery-waste-based fertilizers on ryegrass, specifically focusing on their impact on nitrogen use efficiency, plant growth, and soil properties. This study intends to show the significant potential for the use of tannery-waste-based fertilizers in improving nitrogen use efficiency in ryegrass cultivation. This work lays the groundwork for further investigations into sustainable and cost-effective agricultural practices, contributing to the broader efforts in moving towards a circular economy. Taking into account the quantitative aspects of global tannery industry production, where only ca. 20% of the feedstock material (bovine shavings) is transferred to the final product, any processing of the waste stream accounting for the remaining 80% is valuable.

## 2. Materials and Methods

A five-month greenhouse study evaluating the effects of various tannery-waste-based fertilizers on ryegrass was performed. In this study, 7 fertilizers were applied, including acid hydrolysates of shavings (containing or not containing chromium), as well as wet-green shavings (alone, with biochar from “wet white” shavings, or with biochar and a consortium of microorganisms), which is the novelty. Yield and nitrogen uptake were checked at monthly intervals to assess total nitrogen consumption and growth dynamics of ryegrass over the harvest. In this study, an innovative approach of utilizing tannery waste shavings by preparing different types of hydrolysates and treating wet-greens was applied. The new organic nitrogen-based fertilizers were evaluated for nitrogen use efficiency.

In a standardized approach to assessing fertilizer functionality, a series of in vitro tests—such as extraction tests simulating soil solution with agents like sodium citrate and water solubility—along with in vivo plant tests, were conducted. The in vivo testing includes germination studies (initial dose–effect evaluations of functional properties), pot experiments, and field trials, where the impact on crop yield is assessed under real-world conditions. This approach facilitates comparison of results with other studies and ensures compliance with fertilizer registration regulations.

The effect of tanned waste fractions on ryegrass growth was examined through pot experiments in a glasshouse located in Gdynia Wiczlino, Northern Poland (54.5048 N, 18.4038 E), from April to September 2022.

### 2.1. Fertilizers

Three types of bovine leather waste were used in the research: (1) Cr containing shavings, (2) shavings free of Cr for hydrolysation, and (3) shavings from wet-greens for blending with biochar from shavings free of Cr and blending with the same biochar and a microbial treatment. The mentioned waste is generated during the processing of tanned leather for upholstery purposes in BADER Polska sp. z o.o., located in Bolesławiec, south-west Poland. The global BADER group covers 20% of the automotive upholstery market worldwide. This waste is generated during the processing of tanned leather for upholstery purposes. To improve the fertilizing properties of the leather waste, a pretreatment process was implemented. A detailed description of the fertilizers and pre-treatments used in tests is presented in Appendix A in the Appendix A. Seven fertilizer materials were investigated in a glasshouse experiment and compared with the commercially available mineral NP fertilizer. Appendix A in the Appendix A show the amounts of tanned leather waste fertilizers based on N content added to the soil. Appendix A in the Appendix A show the elemental composition of the materials. The pretreatment involved both hydrolysing the shavings with a mixture of acids (below) and amending the wet-greens with biochar derived from wet-whites (shavings free of Cr) and additionally with a commercial microbial consortium (BactoFos, manufactured by Bacto-Tech, Toruń, Poland). Appendix A were added to the Appendix A with an elemental composition of (1) biochar from shavings FOC, which was used for OLIVb and OLIVb_bio treatments, (2) HFOC, and (3) HC. The composition of wet-greens used in OLIV, OLIVb, and OLIVb_bio treatments is in possession of the manufacturers. Below, the wet-green tanning agents used and their manufacturers (in brackets) are listed: TRUPOSLIP P (Trumpler), TRUKALIN CAN (Trumpler), SELLATAN RL LIQ (TFL), OROPON GRAN (TFL), Optitian TD 626L (SMIT), Neopalin PF (Dr. Eberle Clever Chemistry), PELLAN GLS (Pulcra Chemicals), wet-green OBE1 (Natural Leather Solutions), Tanicor BN (Stahl), and BAYKANOL SL (LANXESS Energizing Chemistry).

HC and HFOC: Shavings were hydrolysed with a blend of acids including phosphoric (V) (12%), fumaric (2%), oxalic (1.7%), and citric (1.7%). To this, 200 g of solution and microelement sulphate salts (Cu, Zn, and Mn) at 0.25% were added to 200 g of shavings and maintained at 100 °C for 1 h. The pH of the resulting hydrolysate was adjusted to 2.5 with potassium hydroxide. The hydrolysate was then granulated by mixing with biomass ash in a ratio of 350 g of the solution to 450 g of ash. EU regulations set a limit for chromium (VI) in organic-mineral fertilizers, allowing no more than 2 mg/kg. Speciation analysis of leather waste fertilizers typically shows chromium (VI) levels below 1 mg/kg. Fertilizers derived from hydrolysates using sulphuric (VI) and phosphoric (V) acids and granulated with biomass ash demonstrate chromium (VI) content within acceptable limits [8,26].

HYDC and HYDFOC. These hydrolysates were obtained while extracting collagen from shavings using strong HCl to maximize the collagen recovery. Thus, they only constitute a byproduct after the collagen extraction. This approach was chosen to test this processed waste as fertilizer without any further pretreatment and compare it with dedicated hydrolysates, described in the previous paragraph. Basic characteristics are shown in the Appendix A in Appendix A. The protein content was based on total nitrogen using the Kjeldahl method according to PN-75/A-04018; the dry matter content was based on the oven method according to AOAC 991.43; the ash content was based on PN-A_88022:1959, using the ash content by weight method consisting of burning samples in quartz crucibles in a gas burner flame and then incinerating them in a muffle furnace at 650 °C; and the fat content was determined by the standard Soxhlet method, extracting with petroleum ether for 6 h, with a v/m ratio of solvent to extracted product of 4:1.

As a reference fertilizer, an NP mineral fertilizer FLOROVIT (MF) was used: 19.0% total N, of which 5.4% nitrate N and 13.6% ammonium N; 6.0% P_2_O_5_ in neutral ammonium citrate and water; 3.9% P_2_O_5_ in water; 2.5% MgO total; and 4.0% Fe total. Total N contents in terms of dry matter of the tanned waste fractions were as follows: HC 11, HFOC 7.5, HYDC 20.37, HYDFOC 18.14, OLIV 63.04, OLIVb 63.04, and OLIVB_bio 63.04 gN/kg. The FLOROVIT standard (known for its quick effect) was selected as a representative mineral fertilizer, commonly available in garden stores. It is understood that phosphorus content in the tannery waste fractions is generally low (<0.5% across most fractions), with the exception of HC, which has over 8% due to additional supplementation.

Assuming the plants would respond to nitrogen-rich organic waste, fertilizer applications ranged from 20 to 370 kg N/ha to achieve the plateau on the nitrogen response curve. Appendix A outlines the experimental design, detailing the fertilizer dosage and the corresponding nitrogen amounts per pot. Starting with the standard ryegrass dose of 20 kg N/ha, as recommended for mineral fertilizers, the dosage was then increased in increments of 50 kg N/ha up to 170 kg N/ha, the maximum annual nitrogen limit for natural fertilizers on Polish farmland (according to the Council of Ministers’ Regulation of 12 July 2018, for reducing water pollution from nitrates in agriculture). To reach the over-fertilization plateau, further increments were added, with the final dose reaching 370 kg N/ha, increased by 100 kg N/ha from the previous level.

### 2.2. Soil and Plants

The plants were cultivated in the <2 mm sieved fraction of sandy soil mixed with peat in a weight ratio of sand–peat = 5:1, equivalent to a 1:1.5 volume ratio. The soil properties included dry matter (d.m.) at 88.77%, organic matter (o.m.) at 6.08%, Total Kjeldahl Nitrogen (TKN) at 1.32 gN/kg d.m., Total Phosphorus (TP) at 185.58 mgP/kg d.m., P-Olsen at 19.04 mgP/kg d.m., Total Potassium (TK) at 610 mgK/kg d.m., K-Olsen at 64.94 mgK/kg d.m., pH 8.287, redox potential 63.8 mV, and electrical conductivity (EC) at 159.7 mS/cm. Approximately 1.75 kg of this prepared soil was placed in each pot with an internal diameter of 14.5 cm (surface area: 0.0165 m^2^). Supplemental nutrient solutions (except for nitrogen) were added according to the following recipe: K_2_SO_4_ (42 g/L) at 12 mL/pot and 6 mL/pot of solutions including CaCl_2_·2H_2_O (90 g/L), MgSO_4_·7H_2_O (24 g/L), MnSO_4_·H_2_O (6 g/L), ZnSO_4_·7H_2_O (5.4 g/L), CuSO_4_·5H_2_O (1.2 g/L), H_3_BO_3_ (0.42 g/L), CoSO_4_·7H_2_O (0.16 g/L), and Na_2_Mo_4_·2H_2_O (0.12 g/L). Soil and nutrients were mixed into the top 5 cm of the soil layer.

A total of 80 ryegrass seeds (0.5 g) consisting of a mix of *Lolium perenne* (40%), *Lolium multiflorum* ‘*Estanzuela 284*’ (20%), *Festuca rubra* (25%), and *Lolium hybridum* (15%) were sown on the soil surface in each pot and covered with an additional 80 g of soil. Experiments were conducted in duplicate, with pots re-randomized weekly to balance light exposure. Soil moisture was maintained at field capacity (20% g H_2_O/g soil d.m., approximately 26.4% cm^3^ H_2_O/cm^3^ soil) with deionized water (DIW). The ryegrass was harvested monthly over four months by cutting the tops about 1 cm above the soil. The harvested plants were then dried in paper bags at 105 °C to a constant weight.

### 2.3. Soil and Plant Analysis

Soil samples were analysed for pH, EC, and redox potential (1:5 H_2_O) before planting and after the last harvest [34]. The phosphorus concentration in liquid samples was measured using a portable spectrophotometer (Hach DR3900, Hach Company, Wrocław, Poland) following the Hach Method 8048, which includes a mineralization step. Prior to analysis, water–soil samples were filtered through a paper filter and then through a 0.45 µm syringe filter. Ryegrass tops from each of the four harvests were dried, ground, and analysed for total nitrogen. Total Kjeldahl Nitrogen (TKN) was measured using the Kjeldahl method, involving digestion (SpeedDigester K-436, Büchi, Uster, Switzerland) in concentrated H_2_SO_4_ with a titanium-based catalyst. This was followed by steam distillation (K-355 distillation unit, Büchi) into a boric acid solution containing Tashiro indicator, then titration with HCl to quantify the released ammonia.

### 2.4. Theory/Calculation

#### 2.4.1. Agronomic Effectiveness

The absolute agronomic effectiveness (AAE) and relative agronomic effectiveness (RAE) of the materials were calculated for each of the four harvests and for the cumulative total at the experiment’s conclusion. These calculations were based on total nitrogen uptake data for each individual harvest and cumulatively (harvests 1–4) as well as cumulative dry matter yield data. The AAE is determined by the slope of the best-fit line between plant nitrogen uptake and the nitrogen application rate, while the RAE is calculated as the ratio of the AAE of each material to the AAE of the reference fertilizer. This is a standard approach for assessing the performance of different fertilizers [35].

#### 2.4.2. Nitrogen Use Efficiency

Nutrient/nitrogen use efficiency (NUE) refers to the ability of crops to take up and utilize nutrients for optimal yields; therefore, the concept involves three major processes in plants: uptake, assimilation, and utilization of nutrients. It is calculated by the difference between nitrogen use in kg N/ha for a given dose and the nitrogen use in kg N/ha for the control scenario (without fertilizer) with reference to the input N with the fertilizer applied (kg N/ha). This parameter is widely used in similar studies [36,37].

### 2.5. Limitations

The research was performed with the following limitations: (1) it was carried out in a glasshouse, (2) using small-diameter pots, (3) under semi-controlled meteorological circumstances, (4) for a limited number of harvests, (5) for one crop, and (6) for one soil type.

## 3. Results

### 3.1. Ryegrass Biomass Yield and Nitrogen Uptake Responses to Applied Waste-Based Fertilizers

#### 3.1.1. Dry Matter Ryegrass Yield

Generally, according to Figure 1 dry grass yields ranged from 1.7 to 4.5 g of dry matter per pot. The maximum yields were reached after 90 days. Only “light” hydrolysates reached the biomass yields comparable to mineral fertilizer after 90 and 120 days, whereas wet-green-based fertilizers remained at a stable plant yield level for most of the time, not exceeding 1.5 g d.m./pot. Biochar-amended wet-green fertilizers provided up to three times better yields than untreated wet-greens, expressing the maximum of growth around dosage 120 kg N/ha. Heavy hydrolysates allowed the growth of biomass later, only after 60 days and further showing maximums for small doses around 70 kg N/ha, with hydrolysates without chromium having a slight advantage. These materials provided similar yields to the wet-green ones for smaller doses and exceeded the light hydrolysate yields after 120 days. They were three times better than all other materials after 150 days. After a prolonged period, heavy hydrolysates without chromium continued to provide yields after 170 kg N/ha. In contrast, those with chromium completely inhibited growth after this dose, irrespective of the growth time.

These findings indicate that hydrochloric acid, used for heavy hydrolysates, likely dissolved many other pollutants into the soil. The soil neutralized part of the unwanted elements, providing some availability of nitrogen to the plants, but only for the first half of the range of doses. For the remaining higher doses, the growth was inhibited.

In general, the findings are in line with the hypothesis that hydrolysates behave similarly to mineral fertilizer after certain time, and organic materials such as wet-green provide smaller yields but in a more stable manner over a longer time, and the plant response is more stable and less driven by the application dose.

#### 3.1.2. Nitrogen Uptake

The nitrogen concentration in plant dry mass varied from 10 to 55 g N/ha. Mineral fertilizer provided the highest N concentrations among all materials only at the beginning of growth. Then, after 60 days, concentrations for MF- and wet-green-fertilized plants were similar, and after 90 and 120 days, they showed the highest amounts for heavy hydrolysates (up to 55 g N/ha) and medium range amounts (ca. 30–50% less) for wet-green ones. The lowest (ca. 60–90% less) and the least dosage-dependent amounts were observed for light hydrolysates and mineral fertilizer. The effect of chromium on nitrogen uptake is unclear. Nonetheless, heavy hydrolysates without chromium provided higher nitrogen concentrations, particularly after 120 kg N/ha, and only in the later growth stages after 90 days.

This is in line with dry matter yields, which started only after 90 days due to nitrogen uptake. Generally, in the literature, the phenomenon observed of fast exploitation of a nitrogen source from highly available mineral fertilizer and light hydrolysates is supported [38,39,40,41,42]. Wet-green-based materials with no clear difference between untreated ones and biochar/biochar-microbe-amended ones show a moderate increase in nitrogen concentration with dosage application; however, it is still much higher than mineral fertilizer and light hydrolysates after 90 days.

### 3.2. Total Ryegrass Growth Dynamics Across Harvests

ThE Figure 2 was created to visualize the distribution of ryegrass total dry matter yield across harvests (as a sum for all doses) to show which materials provided the highest yields in a given month of growth. For mineral fertilizer and light hydrolysates, most of the biomass is generated in month 2 and 3 (70–75%), whereas the rest is formed in months 1 and 4 (25–30%). In total, mineral fertilizer provided 48 g d.m./pot, and light hydrolysates provided 30 g d.m./pot. In contrast, wet-greens provide more gradual growth of biomass over the first three months, providing ca. 20–35% of total biomass each month. Wet-greens still provide up to 20% of total biomass in the last month, which is two times more than mineral fertilizer in this month. It seems that the presence of biochar amendment in this treatment shifts the biomass growth to later months as the maximum biomass growth for month 1 was only observed for untreated wet-greens (38% of total biomass). Biochar amendments and biochar amendments with microbial treatment also provided 60–110% higher yields than untreated wet-greens, especially after months 2, 3, and 4. In total, during the first 3 months, wet-greens provided 3.6–8 g d.m./pot vs. mineral fertilizer and light hydrolysates, providing up to three times more during months 2 and 3 (10.5–20.8 g d.m./pot). No biomass increase was found in month 1 for the heavy hydrolysates, where most of the growth (up to 50% of the total biomass) was found only in month 4, especially for the chromium-free hydrolysate. In absolute terms, this was from 22 to 23 times higher than the control treatment but still lower than most of the other materials in this time of growth. The Figure 2 also visualizes the differences in dry matter distribution dynamics between harvests in absolute terms. The total dry matter yield in g d.m./pot presented from the highest to the lowest values is as follows: MF 48.08, HC 29.93, HFOC 28.93, OLIVB 24.62, OLIVB_bio 23.60, OLIV 17.17, CONTROL 10.4, HYDFOC 7.50, and HYDC 6.50.

### 3.3. Nitrogen Utilization by Ryegrass

Figure 3 shows nitrogen utilization expressed as total nitrogen extracted by ryegrass per hectare. In general, the highest amounts of nitrogen (up to 80 kg N/ha) are utilized by plants fertilized with mineral fertilizer after 60 days of growth, and they decrease dramatically after 120 days to mostly values less than 10 kg N/ha. This is in opposition to studied materials that express N utilization in a more gradual manner up to 30 kg N/ha across the first three months followed by a decrease to under 20 kg N/ha after the last month.

The dynamics of N utilization across growth time are as follows. First, wet-greens express up to 2–5 times better N utilization than light hydrolysates after 30 and 120 days. This is especially true for the amendment with microbially treated biochar. After 60 and 90 days, the N utilization of all studied materials, apart from heavy hydrolysates and untreated wet-greens, is more or less the same. Second, chromium-containing light hydrolysates show a maximum of N utilization at 60 days at 220 kg N/ha and for 90 days at 270 kg N/ha. Third, after 90 days, heavy hydrolysates start showing N utilization up to 20 kg N/ha. Compared to others, they still express very high N utilization for small doses (less than 170 kg N/ha) after 120 days. This is up to two times higher than biochar-amended wet-greens and up to 10 times higher than light hydrolysates, mineral fertilizer, and untreated wet-greens. Fourth, microbiologically treated biochar amendments of wet-greens express the highest N utilization among all wet-greens, especially after a dose of 120 kg N/ha after 120 days.

Where the linear regression model was possible to apply, the trend lines were shown with correlation factors and equations on the plots. Absolute agronomic effectiveness (AAE) is defined by the slope of the best-fit line in a linear regression model that represents the response of plant growth to fertilizer dosage. This response can be based on either nitrogen uptake or dry matter yield in relation to the fertilizer dosage. Essentially, nitrogen-based AAE indicates the percentage of nitrogen absorbed by the plant compared to the total nitrogen applied through increasing fertilizer doses. Calculating AAE is a widely used method in agricultural research for evaluating fertilizer performance and is frequently cited in the scientific literature [43,44].

### 3.4. Total Nitrogen Utilization Dynamics Across Harvests

Figure 4 was created to best visualize nitrogen utilization dynamics across months of growth (presented as the sum of all doses’ N utilizations in each month). It shows that control treatment provides up to 60% of nitrogen utilization very early, already after 30 days, whereas mineral fertilizer represents 30% in the same time and 45% of nitrogen utilized later from 30 to 60 days. Nitrogen utilization from light hydrolysates is more evenly distributed over time, with maximums (35–40%) peaking between 60 and 90 days. The difference in nitrogen utilization between chromium-containing light hydrolysates and those without chromium is insignificant, slightly shifting the N utilization more to month 3 for the non-chromium ones. In general, light hydrolysates behave similarly to mineral fertilizer in terms of fast exploitation of the majority of nitrogen during first 3 months; however, in the last month, these materials still provide up to 10% of N utilization vs. 3% for the mineral fertilizer. Biochar amendments of wet-greens shift the nitrogen utilization further to the last three months (70% vs. 58% for pure wet-greens) without significant differences between biochar amendment alone and this amendment after microbial treatment. Nitrogen utilization from soils amended with heavy hydrolysates is strongly retarded due to potential toxicity resulting from strong hydrolysis of many pollutants with HCl. It only shows some measurable numbers after 90 days (35–40%), with the greatest values after 120 days (60–65%) and even after 150 days (only shown on Figure 5). In absolute terms, after month 4, they provided up to three times more nitrogen than mineral fertilizer, two times more than light hydrolysates, and up to 10% more than biochar-amended wet-greens. Heavy hydrolysates without chromium provided around 10% more nitrogen utilization after month 3 and up to 80% more nitrogen after month 4 compared to chromium-containing heavy hydrolysates.

The Figure 4 also visualizes the dynamics of the cumulative value of nitrogen utilization for each harvest. The total N utilization in kg N/ha presented from the highest to the lowest values is as follows: MF 717.10, OLIVB 403.84, OLIVB_bio 348.00, HC 291.26, HFOC 264.14, OLIV 263.18, HYDFOC 108.29, CONTROL 106.2, and HYDC 68.88. This order is almost in line with the order of total dry matter yields discussed under Figure 2 (with the advantage of biochar-amended wet-greens being before light hydrolysates in N utilization), evidencing nitrogen being the main growth stimulating factor.

### 3.5. Ryegrass Yield, Nitrogen Uptake, and Nitrogen Utilization After Additional Harvest

Because heavy hydrolysates started showing both dry matter yields and nitrogen utilization after 90 and 120 days rapidly, it was decided to keep the growth for these materials for one more month and compare it with the growth using light hydrolysates. Figure 5 shows three different parameters starting from plant dry matter yield, then nitrogen concentration in plants, and finally nitrogen utilization after 150 days of growth. Heavy hydrolysates express maximums for all parameters visible for small doses ranging below 120 kg N/ha. The only exception is chromium-free heavy hydrolysate, which shows the maximum nitrogen concentration for 220 kg N/ha. Plant dry matter yields are up to seven times higher, nitrogen concentrations up to three times higher, and nitrogen utilization up to seven times higher than scenarios with light hydrolysates. The trend of having better parameters for non-chromium heavy hydrolysates is maintained. It seems like all three parameters follow a similar character for mineral fertilizer and light hydrolysates, and they are less fertilizer-dosage-dependent—only for nitrogen uptake and utilization. Chromium-containing heavy hydrolysates provide growth parameters only until a dose of 120 kg N/ha. After that dose, the growth is strongly inhibited by some unidentified pollutants. Another comprehensive study is currently underway to verify the exact compounds responsible for that. The overall conclusion is that small doses of heavy hydrolysates shift the growth parameters to the third and fourth month and later to the fifth, with an increasing tendency, proving high soil self-purification efficiency and mineralization capability of compounds that are initially unavailable to the plants and potentially toxic compounds that inhibit growth after months 1 and 2.

### 3.6. Cumulative Ryegrass Yield and Nitrogen Utilization

Figure 6 summarizes all the data from Figure 1 (left), referring to plant dry matter yield, and from Figure 3, referring to nitrogen utilization. The cumulative values of plant yield show maximums (4–4.5 g d.m./pot) of dry matter yield for light hydrolysates around 170 to 270 kg N/ha, being still ca. 20% lower than mineral-fertilizer- provided plant yields. Then, the response of the plant to wet-greens is up to 3 g d.m./pot, being more stable across doses, showing maximums for smaller doses (<170 kg N/ha), and making the microbially treated biochar amendments up to 20% better than amendments with only biochar, with pure wet-greens being in last place (<2 g d.m./pot). Heavy hydrolysates generate minimum amounts of dry matter only for small doses (<120 kg N/ha), as mentioned earlier. The cumulative values of nitrogen utilization for waste-based materials are rather similar apart from heavy hydrolysates and untreated wet-greens, which show clearly lower nitrogen utilization. Where it was possible, a linear regression model was applied to show minor differences in the response of N utilization in a mathematical way. Please note that correlation coefficients higher than 0.87 were mostly found, with the single exception of 0.67 for OLIVB_bio. The regression for chromium-containing light hydrolysate was not displayed due to non-linear trend. These values will be used to further calculate relative agronomic effectiveness.

### 3.7. Relative Agronomic Effectiveness and Nutrients Use Efficiency

Figure 7 summarizes all previously presented data on Figure 3, Figure 5 and Figure 6 by showing the calculated relative agronomic effectiveness (RAE) for the fertilizer materials, where the calculation of AAE was possible. RAE is essentially the AAE compared to the AAE of the reference mineral fertilizer (MF). The RAE based on the utilization of N (RAE(N)) was calculated and presented for each subsequent harvest and also as a total value after adding all harvest data. However, the d.m. yield-based RAE (Y) was calculated only as a total value. The N-utilization-based RAE(N) better characterizes the fertilizer material (according to authors’ opinion) as it also contains the dry matter yield when calculating the total N utilization per area. The RAE values for processed waste-based materials can be determined only for the regressions (Figure 3, Figure 5 and Figure 6) with relatively good correlation coefficient, e.g., R^2^ > 0.85.

Figure 7 shows relative agronomic effectiveness data based on the linear regressions of nitrogen utilization data from Figure 3, Figure 5 (right), and Figure 6 (right). Data from 2021 study are also shown, where S.C. denotes Cr containing shavings from the preprocessing, whereas OFOC—free of Cr off-cuts from the post processing [45]. The highest values were noticed after 120 days, reaching 100% and even more for chromium-free light hydrolysates and biochar-amended wet-greens, respectively. Microbially treated biochar amendments of wet-greens exceeded 150% after 120 days, followed by biochar amendments of wet-greens after 120 days, reaching almost 120% and chromium-free light hydrolysates reaching almost 120% after 150 days. Chromium-containing light hydrolysates were not that efficient and did not exceed 53%. Chromium-free light hydrolysates were more consistent in keeping the RAE close to 100% over the time of growth than biochar amendments of wet-greens, which apart from 120 days growth did not exceed 43%. Microbial treatment with the BactoFos product of biochar-amended wet-greens provided up to 30% better RAE than biochar amendments alone. Excluding the RAE values in particular months of growth, their cumulative values for 4 months for the studied materials did not exceed 34%, with biochar-amended wet-greens having the highest values, followed by chromium-free light hydrolysates.

Comparing the data from 2022 to data for some of the materials studied in 2021 (chromium-containing light hydrolysates, ground bovine shavings, and ground splits and offcuts) it seems like chromium-containing light hydrolysates in 2021 provided much better RAE, reaching 95% for 30 days compared to data for 2022 that were two times poorer, reaching as low as 27% for 150 days. These inconsistent results may be due to the heterogeneous structure of the different feedstock materials used by the upholstery industry between 2021 and 2022, or possibly related to the sampling process. As demonstrated in a previous study [45], only mechanical and effective microbial treatment of the leather waste fractions has been found to increase RAE by no more than 10%.

It is important to highlight that relative agronomic effectiveness tries to capture the fertilizer performance taking into account the response of nitrogen utilization to fertilizer dose across all applied doses and calculates a single output value information for the whole trend. However, nutrient use efficiency (NUE) represents a simpler approach that only shows how much nitrogen is extracted from the agricultural system for the given fertilizer material and given dose compared to the control treatment. This is why NUE is calculated for the single dose separately, unlike RAE, which tries to capture the whole trend. These differences in approach enable a comprehensive understanding of the fertilizer performance of alternative waste-based materials.

Figure 8 and Table 1 and Table 2 show NUE values for all the studied materials and a reference material for the singular dose in each month of growth and finally the total NUE after the whole experiment (4 months). The colours from dark red to dark green help to visualize the levels of NUE. The highest values were observed for chromium-free heavy hydrolysates, reaching 51.5% after 60 days. However, considering previous observations, these materials perform only at small doses and during later months, which renders their use impractical and potentially toxic. This is also backed up by looking at total NUE, where the values did not exceed 32% for chromium-free and 20% for chromium-containing ones. Excluding heavy hydrolysates, the best fertilizer material identified was microbially treated biochar amendment of wet-greens, reaching 37% after 30 days and in total even over 100% for the smallest dose (20 kg N/ha). These microbial products provided up to 30–50% more total NUE than biochar amendment of wet-greens alone, but only for the small dose (20 kg N/ha). For higher doses, this microbial treatment effect was less visible, reaching a few percent in particular months of growth and up to 20% in total. Pure untreated wet-greens showed the smallest NUE values, not exceeding 7% in each harvest and not even 20% in total. However, because according to the product data sheet, most of the wet-green tanning agents pose a risk to the aquatic environment, the thorough examination of the final fertilizer composition from wet-greens should be carried out prior to soil application and based upon the national regulation standards. These agents may also have negative effects on animals, but according to product data sheets, this was only noticed at a concentration >5%. Moreover, the 70–100% biodegradability is reached within 1 month, which is a positive aspect of using such material in the environment. Light hydrolysates showed up to 25% of NUE for the advantage of chromium-free ones and only after 60 days but in total up to 40%, and the total N use efficiency was slowly decreasing over fertilizer dose. The NUE for the mineral fertilizer exceeded 35% for the single month (60 days) and reached 54% in total for 220 kg N/ha. In general, total NUE values were decreasing with increasing fertilizer dose for the alternative materials, whereas for the reference one, the maximum was around 200 kg N/ha. These also give practical information on optimal use of waste materials as fertilizers, providing the highest N use efficiencies with the smallest possible amounts of waste being applied on land.

### 3.8. Residual Soil Properties After 120 Days of Ryegrass Growth

Figure 9 shows basic soil physical properties after termination of the 120-day glasshouse experiment. For most materials, pH slightly decreased from ca. 8.0 to 7.4 for Cr-free heavy hydrolysates. Biochar-amended wet-greens kept pH at a rather stable level, although biochar is rather alkaline; however, their dose was not high enough to alter the pH. The highest pH reaching 8.8 was observed for Cr-containing light hydrolysate. Soil EC was altered 2–3 times after the application of biochar-amended wet-greens and quite considerably after application of Cr-containing heavy hydrolysates (almost four times more) and significantly after application of Cr-free heavy hydrolysates (6.5 times more). This is due to usage of hydrochloric acid as a very strong reagent hydrolysing many ions to the soil solution. The redox potential of the soil solution, also known as the Eh (oxidation–reduction potential), refers to the tendency of the soil to either gain or lose electrons in its chemical reactions. It is a measure of the soil’s ability to undergo oxidation (loss of electrons) or reduction (gain of electrons) processes. Application of the tested waste materials mostly decreased the redox potential of the soil solution with the increased application rate from a maximum of 220 to 145 (mV), meaning the soil was in a reduced state and turning into more anaerobic conditions. In a reduced soil environment, the availability of oxygen is limited, as it must have been consumed by soil microflora when adding more and more fertilizer to the soil. The low redox potential in the soil can have significant implications for various biogeochemical processes. For example, (1) plant growth: some plants may not thrive or may even die in anaerobic conditions due to the lack of oxygen necessary for their root respiration; (2) nutrient availability: the availability of certain nutrients may be influenced by redox conditions; for instance, in reduced soils, iron and manganese may become more soluble and available to plants, while under oxidized conditions, these nutrients may precipitate and become less accessible; (3) decomposition: the breakdown of organic matter can be slowed down in anaerobic conditions, leading to the accumulation of organic residues in the soil; (4) greenhouse gas emissions: anaerobic conditions in soils can promote the production of greenhouse gases like methane (CH_4_) and nitrous oxide (N_2_O); (5) contaminant mobility: redox conditions can affect the mobility and bioavailability of various contaminants, such as heavy metals and organic pollutants.

European Union regulations on chromium in fertilizers set limits only for Cr (VI), while Cr (III) remains unrestricted, as it is generally considered safe based on the scientific literature. Current regulations are expected to maintain a maximum allowable Cr (VI) content of 2 mg/kg, with no limit on Cr (III) (as outlined in the EU Fertilizer Regulation (EC) 2003/2003). Scientific studies suggest a safe threshold for Cr (III) of up to 5000 mg/kg.

## 4. Discussion

Leather waste is a very valuable source of nitrogen, a key element for proper plant growth [46]. It is worth noting that the use of nitrogen mineral fertilizers causes rapid dissolution of granules and fast release of elements. Niedziński et al. (2021) in their work noted that within 35 days, 70% of nitrogen is released from ammonium granules, while urea granules release as much as 98% [47]. In the case of organic fertilizers, this value ranges from 15 to 28%, significantly reducing potential emissions in the form of ammonia and NOx [47]. However, direct application of waste from the tanning industry as organic preparation does not allow efficient use of nitrogen, due to the presence of nutrients in forms not available to plants [48]. The use of untreated wastes makes it possible to increase yields, but the process of natural decomposition and increased nutrient availability takes a long time. This paper proposes two main treatment methods, hydrolysis and carbonization with a microbial treatment, as safe technologies for processing waste for fertilizer purposes. The developed processes do not generate by-products, thus fitting closely with the assumptions of a closed-loop economy and green production. Acid hydrolysis leads to the production of short peptides and free amino acids, thus increasing the market value of the proposed products [49]. The use of sulphuric (VI) and phosphoric (V) acids additionally leads to the valorization of fertilizer materials in phosphorus and sulphur, which play a key role in proper plant growth [50,51].

Carbonization of waste tends to produce materials rich in nitrogen and carbon [52]. The process is a very good alternative to combustion, allowing safe processing, eliminating nitrogen emissions in the form of NOx into the air [53]. Biochar has an extremely beneficial effect on crop production, increasing the sorption capacity of the soil and regulating its pH, as well as reducing the leaching of nutrients into ground and surface water [54]. In the case of chromium-containing waste, its use is limited, but it has been proven that the chromium present in biochar from chromium materials is in the highly stable form of chromium carbide, which is not available to plants, and is therefore safe for the environment [55].

Particularly noteworthy is the designed combination with an introduced mixture of live soil bacteria (BactoFos), which naturally decompose phosphates contained in the substrate. The increase in the concentration of available phosphorus has a beneficial effect on the proper development of plants, supporting their proper rooting [56]. The proposed composition naturally increases crop yield with preservation of plant quality and a reduction in carbon footprint by reducing the amount of applied chemicals, similar to organic farming [57].

The key problem that arises with the use of waste in fertilizer production is the presence of chromium from the tanning medium. The element, entering the soil and water, is easily incorporated into the digestive pathway and causes toxic effects on plants as well as many diseases in animals and humans [58]. According to the EU Regulation, only the Cr(VI) content is limited (<2 mg/kg s.m.). Given this, there are no legal barriers to using fertilizers containing higher Cr(III) contents, but due to the possibility of oxidation of Cr(III) to Cr(VI), care must be taken [59]. The safe solution is to use this type of fertilizer in closed (greenhouse) crops, where it is possible to control the transfer of the element into the environment (avoiding potential accumulation in the soil or leaching into groundwater). Controlled plantations using Cr-containing formulations would enable the production of specialty foods (functional hypoglycaemic foods) [60,61]. Also, not all wet-green tanning agents are safe for the aquatic environment; therefore, a thorough examination of the final fertilizer composition from wet-greens should be performed prior to soil application and based upon the national regulation standards.

The positive aspect in the case of leather production is the slow abandonment of treating materials with Cr-containing substances and replacing them with natural tannins in the form of oils [62]. This allows the waste to be safely processed into fertilizers that meet EU requirements with high efficiency and a safe environmental aspect. Recycling this type of waste can replace inorganic fertilizers, reducing greenhouse gas emissions and carbon footprint.

The results of this study could also have applications for other crops not designated for direct consumption, such as grasses, energy crops, flowers, trees, maize, ocra, and pine [10,63,64]. Light hydrolysates could possibly be used for vegetables, rice, etc., simply because the risk of carrying toxic compounds is very low. Heavy hydrolysates after collagen extraction, due to the harmful effects on ryegrass growth in the first 3 months, should not be used for any of these cases, unless additionally pretreated. Wet-green-based materials, due to potential harmful effects in aquatic environments, should be thoroughly examined prior to use at a larger scale; however, their use in controlled conditions, e.g., in greenhouses, could find an application. This is because the negative effects on animals were only noticed at concentration >5%, and the 70–100% biodegradability is reached within 1 month.

Quantitatively speaking, our industrial partner globally covers 19% of world car upholstery production (23% in Europe). The above solution could help managing the substantial part of the waste stream amounting for 90 t/day globally resulting from 116,000 m^2^ of input bovine leather material a day. Assuming 40% of this amount as bovine shavings from the feedstock material (both Cr-containing and Cr-free) and the amount of nitrogen reaching 10% (*w*/*w*) and average NUE (for both light hydrolysates and microbially altered and biochar amended wet-greens) being ca. 70%, that gives 9198 t of recycled nitrogen per year, which is returned to the environment as plant biomass.

On top of sustainability aspects, any transformation of the tanned leather waste fractions into the secondary products to be implemented in agriculture is worth investigating as nowadays the utilization costs of leather waste are within 500–1000 EUR/tonne.

Future research directions could be centred around validation of toxicology aspects when heavier waste is used; further microbially induced nutrients’ bioavailability stimulation, effects on growth and N uptake over longer periods of time; speciation of N forms and distribution in soil, roots, and plant fractions; applying other cropping systems and under different climatic conditions; and optimizing collagen extraction while maintaining safer extracts and byproducts which can be directed back to the soil.

## 5. Conclusions

Seven differently treated tanned leather waste fractions were tested for ryegrass growth as alternative fertilizers. Although plant dry matter yields were generally better for light hydrolysates, the final findings on the nitrogen use efficiency show that microbially incubated, biochar-amended wet-greens provided the highest nitrogen use efficiencies, exceeding 100% after 4 months of growth (for 20 kg N/ha) and varying from 17% to 37% in particular months. This is backed up by another parameter (relative agronomic effectiveness) that for these materials exceeded 150% for a single month and in total was around 33%. Biochar amendments significantly increased agronomic parameters for wet-greens, and their microbial treatment enhanced them even further. Light hydrolysates also exhibited quite good performance as the relative economic effectiveness reached 118% for a single month, and nutrient use efficiency reached in total 39%. In spite of higher nitrogen concentrations in harvested plant tops and consequently higher NUE values for heavy hydrolysates (up to 51.5%), they totally inhibited the growth after the 170 kg N/ha dose, especially the ones with Cr. In general, residual soil physical properties were not affected considerably, apart from treatments with heavy hydrolysates that increased the soil electrical conductivity substantially.

Summing up, although good availability of nutrients in the first stage of growth from hydrolysates is a good point when farmers count on fast effect, this research shows that minimally treated wet-greens with no Cr provide even better results, at the same time avoiding additional chemicals (acids) used for treatment. This helps keep the carbon footprint of the whole process as low as possible while contributing to the real principle of a circular economy where no additional chemicals are introduced to the system. It is further recommended to verify the very long-term effect of such waste usage on the ryegrass growth, N uptake, and soil properties, e.g., over the full year vegetation period including winter. The soil once fertilized in the first season would probably hold part of the nutrients in the waste for subsequent utilization in the second season, without the need for re-fertilization. The next dosages could then be applied in the third season and every second season thereafter. This approach would have given the full picture of tanned waste fractions’ usage as N fertilizers [35].

Processed leather waste fractions show promise as organic nitrogen fertilizers, offering an opportunity to repurpose a problematic waste stream while providing essential nutrients for plant growth. This was observed especially for the microbially treated biochar amendment of wet-greens reaching 37% after 30 days and in total even over 100% for the smallest dose (20 kg N/ha). However, further research is needed to optimize their use, address environmental concerns, and develop guidelines for their safe and effective application. Integrating processed leather waste fractions into agricultural practices could contribute to sustainable waste management and enhance soil fertility, fostering a more circular and environmentally friendly approach to agriculture.

## Figures and Tables

**Figure 1 materials-17-05741-f001:**
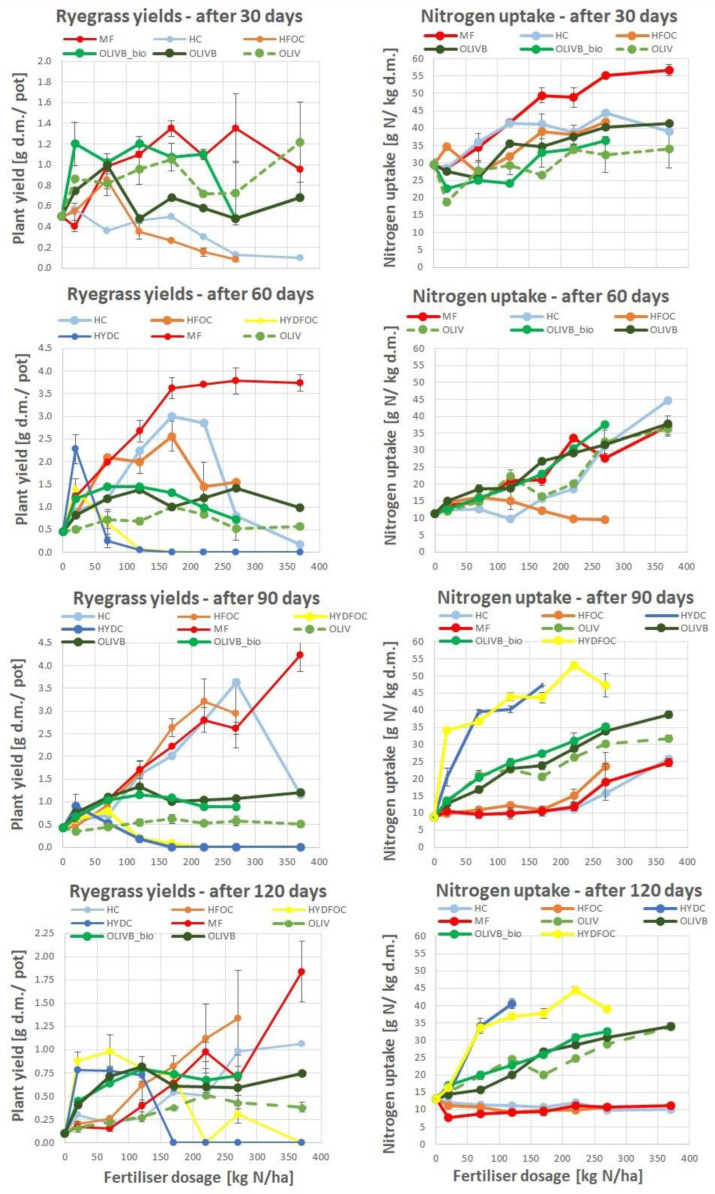
Ryegrass biomass yields response (**left**) and nitrogen uptake (**right**) to tanned-leather-waste-based fertilizers after 4 subsequent harvests, as compared to the mineral fertilizer (MF) for harvests after 30, 60, 90, and 120 days.

**Figure 2 materials-17-05741-f002:**
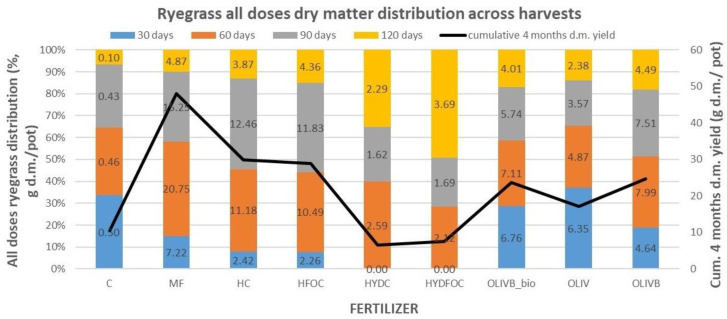
Ryegrass growth (as total dry matter yield) dynamics across four harvests for 7 treatments, mineral fertilizer, and a control: percentage of total dry matter and absolute values in each harvest (**bars**) and total cumulative dry matter in all the harvests (**black line**).

**Figure 3 materials-17-05741-f003:**
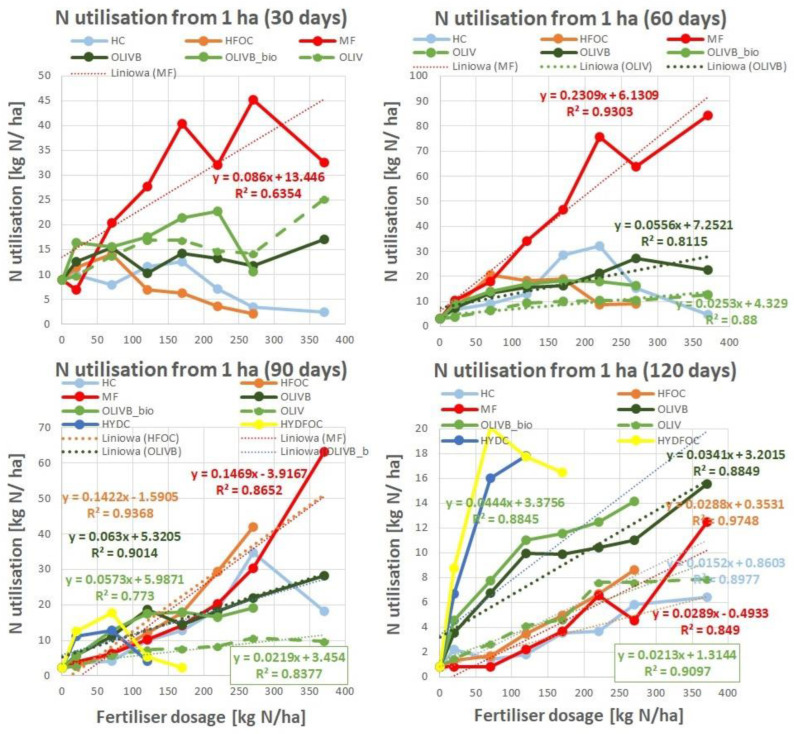
Nitrogen utilization by ryegrass grown on soil amended with tanned-leather-waste-based fertilizers after 4 subsequent harvests, as compared to the mineral fertilizer (MF) for harvests after 30, 60, 90, and 120 days.

**Figure 4 materials-17-05741-f004:**
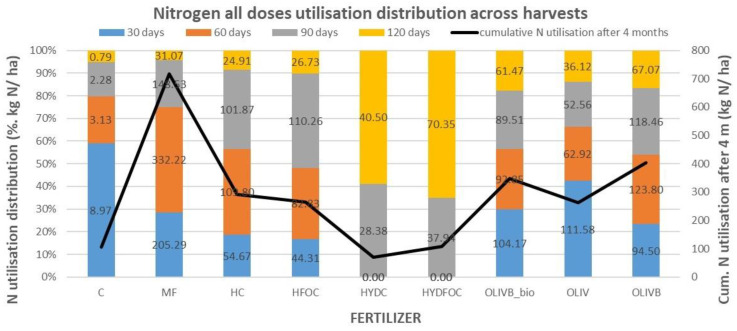
Nitrogen utilization (as total amount per single treatment in kg/ha) dynamics across four harvests for 7 treatments, mineral fertilizer, and a control: percentage of N utilization in each harvest and absolute values (**bars**) and total cumulative N utilization in all the harvests (**black line**).

**Figure 5 materials-17-05741-f005:**
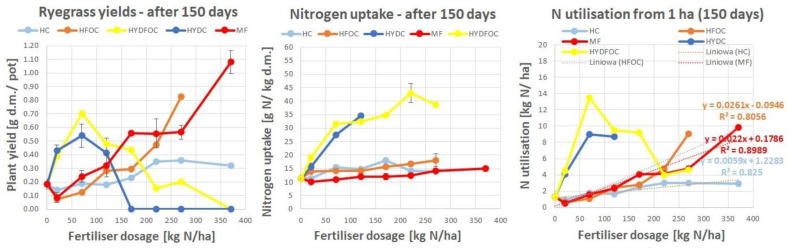
Ryegrass dry matter yields (**left**), nitrogen uptake (**middle**), and nitrogen utilization (**right**) after an additional harvest (150 days) for 4 chosen treatments and mineral fertilizer.

**Figure 6 materials-17-05741-f006:**
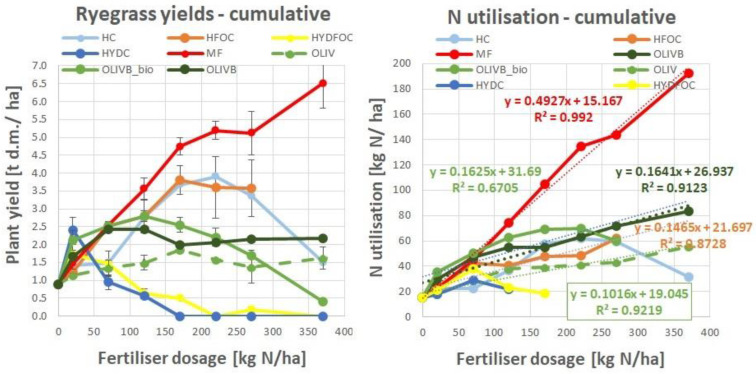
Cumulative ryegrass dry matter yields (**left**) and nitrogen utilization (**right**) for 7 chosen treatments and mineral fertilizer.

**Figure 7 materials-17-05741-f007:**
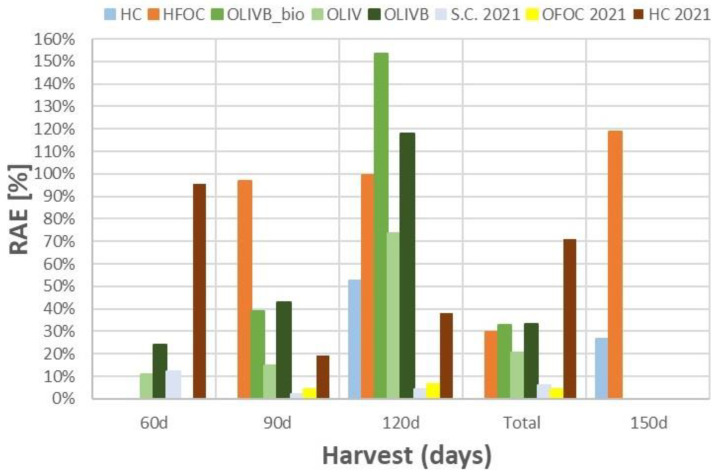
Relative agronomic effectiveness comparison for the processed leather waste fractions across harvests includingdata from 2021.

**Figure 8 materials-17-05741-f008:**
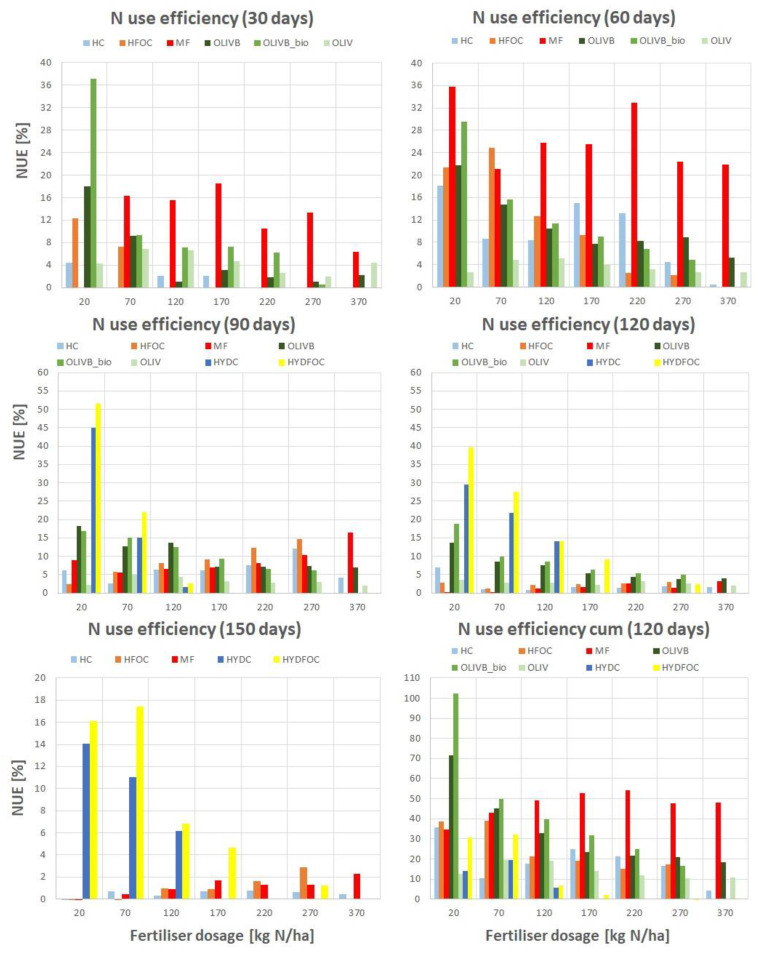
Nitrogen use efficiency (NUE) for the processed leather waste fractions across harvests (from 30 to 150 days) and cumulative values after 120 days.

**Figure 9 materials-17-05741-f009:**
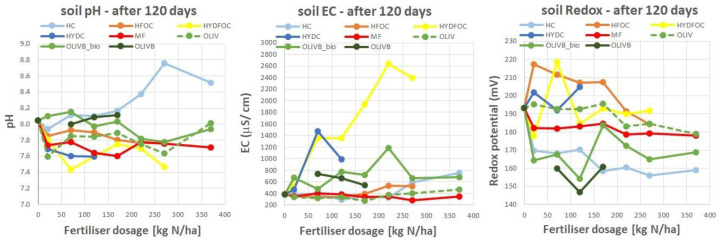
Soil physical properties after 120 days of growth.

**Table 1 materials-17-05741-t001:** Nitrogen use efficiency (NUE) for the processed leather waste fractions across harvests (after 30, 60 and 150 days). NA—data Not Available. Conditional formatting is applied to better visualise the levels of NUE across treatments and doses. The most red—the lowest values, the greenest—the highest values, yellow and orange—mid values.

NUE (N Use Efficiency) 30 days, %								
N dose	MF	HC	HFOC	HYDC	HYDFOC	OLIVB_bio	OLIV	OLIVB
[kg/ha]	%							
20	−9.95	4.44	12.27	−44.86	−44.86	37.13	4.28	18.04
70	16.34	−1.61	7.22	NA	NA	9.39	6.91	9.25
120	15.56	2.15	−1.74	NA	NA	7.13	6.62	1.12
170	18.48	2.07	−1.60	NA	NA	7.29	4.64	3.12
220	10.51	−0.87	−2.45	NA	NA	6.26	2.61	1.90
270	13.40	−2.03	−2.52	NA	NA	0.60	1.95	1.01
370	6.39	−1.79	NA	NA	NA	NA	4.39	2.17
**NUE (N Use Efficiency) 90 days, %**								
20	8.88	6.26	2.36	44.99	51.53	16.85	2.19	18.24
70	5.64	2.59	5.74	15.08	21.95	15.08	4.92	12.76
120	6.64	6.38	8.23	1.67	2.66	12.59	4.39	13.68
170	7.03	6.15	9.04	NA	NA	9.26	3.23	7.13
220	8.14	7.48	12.34	NA	NA	6.52	2.81	7.18
270	10.38	12.04	14.72	NA	NA	6.20	3.09	7.28
370	16.50	4.27	NA	NA	NA	NA	2.02	7.02
**NUE (N Use Efficiency) 150 days, %**								
20	−3.76	−1.51	−3.19	14.09	16.13	NA		
70	0.48	0.72	-0.28	11.06	17.44			
120	0.89	0.29	0.94	6.19	6.83			
170	1.66	0.73	0.90	NA	4.66			
220	1.32	0.78	1.61	NA	NA			
270	1.32	0.66	2.88	NA	1.26			
370	2.32	0.45	NA	NA	NA			

**Table 2 materials-17-05741-t002:** Nitrogen use efficiency (NUE) for the processed leather waste fractions across harvests (after 60, 120 and cumulative after 120 days). NA—data Not Available. Conditional formatting is applied to better visualise the levels of NUE across treatments and doses. The most red—the lowest values, the greenest—the highest values, yellow and orange—mid values. For the cumulative values, the ones higher than 50% were marked bold.

NUE (N Use Efficiency) 60 days, %							
MF	HC	HFOC	HYDC	HYDFOC	OLIVB_bio	OLIV	OLIVB
%							
35.78	18.17	21.34	-15.66	-15.66	29.53	2.70	21.77
21.06	8.60	24.88	NA	NA	15.61	4.81	14.78
25.73	8.36	12.69	NA	NA	11.41	5.17	10.51
25.53	14.97	9.28	NA	NA	8.99	4.01	7.73
32.96	13.20	2.51	NA	NA	6.81	3.23	8.29
22.42	4.54	2.16	NA	NA	4.92	2.71	8.85
21.92	0.40	0.00	NA	NA	NA	2.61	5.21
**NUE (N Use Efficiency) 120 days, %**							
0.03	7.05	2.81	29.60	39.68	18.92	3.54	13.70
0.05	0.93	1.27	21.76	27.53	9.91	2.73	8.55
1.18	0.86	2.22	14.16	14.15	8.51	2.81	7.64
1.70	1.61	2.44	NA	9.23	6.32	2.26	5.33
2.62	1.33	2.68	NA	NA	5.31	3.12	4.39
1.40	1.86	2.90	NA	2.42	4.95	2.52	3.79
3.16	1.53	0.00	NA	NA	NA	1.92	3.98
**NUE (N Use Efficiency) TOTAL 120 days, %**							
34.74	35.92	38.79	14.07	30.69	**102.43**	12.70	71.75
43.09	10.50	39.12	19.54	32.19	**50.00**	19.36	45.34
49.11	17.74	21.39	5.74	6.73	39.65	18.99	32.96
**52.74**	24.80	19.15	NA	2.09	31.86	14.13	23.32
**54.23**	21.13	15.07	NA	NA	24.90	11.77	21.76
47.59	16.41	17.26	NA	-2.91	16.67	10.27	20.94
47.96	4.41	NA	NA	NA	NA	10.94	18.38

## Data Availability

The raw data supporting the conclusions of this article will be made available by the authors on request.

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
