# Peer review of "Leather Waste Hydrolysation, Carbonization, and Microbial Treatment for Nitrogen Recovery by Ryegrass Cultivation"

_materials, 2024, doi:10.3390/ma17235741_

Round 1
Reviewer 1 Report
Comments and Suggestions for Authors
This article is based on tannery waste and is used as a fertiliser to support the circular economy. Related to the structure of the article, some suggestions are recommended.
1. The title does not reflect the work of the author(s). Could they keep it informative so that the reader may not be confused?
2. The abstract is too extensive. Could they condense their idea and be concise in their writing?
3. The introduction is unnecessarily extended. Some paragraphs are not properly cited.
4. The result section is okay, but the quality of the figures is not good.
In a nutshell, I would strongly recommend to curtail the length of this article.
Comments on the Quality of English Language
Check the missing prepositions and articles.
Author Response
Reviewer 1
This article is based on tannery waste and is used as a fertiliser to support the circular economy. Related to the structure of the article, some suggestions are recommended.
- The title does not reflect the work of the author(s). Could they keep it informative so that the reader may not be confused?
Answer: Thank you for this comment. The new title suggestion is: “Leather waste hydrolyzation, carbonization and microbial treatment for nitrogen recovery by ryegrass cultivation.”
- The abstract is too extensive. Could they condense their idea and be concise in their writing?
Answer: Thank you for this comment. The Abstract has been shortened.
- The introduction is unnecessarily extended. Some paragraphs are not properly cited.
Answer: Thank you for this comment. The Introduction is 1.75 page long, I have shortened it slightly, improved the paragraphs and improved the citations.
- The result section is okay, but the quality of the figures is not good.
Answer: Thank you for this comment. The quality of Figs. is good enough in the separately attached files but when the PDF is formed, the quality gets worse.
- In a nutshell, I would strongly recommend to curtail the length of this article.
Answer: Thank you for this comment. We appreciate it. The article has been shortened. Tables 1 and 2 were moved to the Supplementary Material.
- Check the missing prepositions and articles.
Answer: Thank you for this comment. The missing references were added to the list and properly cited.
Reviewer 2 Report
Comments and Suggestions for Authors
Manuscript is about an interesting topic that incides on the circular economy and recycling, that is important in order to get sustainabilitiy and major efficiency in process. Although The reuse of wastes to get a benefit in another area is a good labor, it is important that authors do an exhaustive materials characterization, overall of the used wastes before and after of any treatment
It is suggested to show chemical characterization about elemental or atomic composition or further experiments about nitrogen content because this section is not so clear to me. Authors could do some scanning electron microscopy experiments or any analytical technique to get information about the elements considered such as X-ray photoelectron spectroscopy or another
Comments on the Quality of English LanguageEnglish could be revised
Author Response
REVIEWER 2
- Manuscript is about an interesting topic that incides on the circular economy and recycling, that is important in order to get sustainability and major efficiency in process. Although The reuse of wastes to get a benefit in another area is a good labor, it is important that authors do an exhaustive materials characterization, overall of the used wastes before and after of any treatment.
Answer: Thank you for this comment. The appropriate discussion has been added to the Materials and Methods and to the Results/ Discussion sections.
Materials and Methods:
“Table S1 was added to the Supplementary Material with elemental composition of (1) biochar from shavings FOC which was used for OLIVb and OLIVb_bio treatments, (2) HFOC and (3) HC. The composition of wet-greens used in OLIV, OLIVb and OLIVb_bio treatments is in possession of the manufacturers. Below the list of wet-green tanning agents and the manufacturers (in brackets): TRUPOSLIP P (Trumpler), TRUKALIN CAN (Trumpler), SELLATAN RL LIQ (TFL), OROPON GRAN (TFL), Optitian TD 626L (SMIT), Neopalin PF (Dr. Eberle Clever Chemistry), PELLAN GLS (Pulcra Chemicals), wet-green OBE1 (Natural Leather Solutions), Tanicor BN (Stahl), BAYKANOL SL (LANXESS Energizing Chemistry). Because most of these agents pose a risk to the aquatic environment, the thorough examination of the final fertilizer composition from wet-greens should be done prior to soil application”.
Results:
“However, because according to the product data sheet, most of the wet-green tanning agents pose a risk to the aquatic environment, the thorough examination of the final fertilizer composition from wet-greens should be done prior to soil application an decided upon the national regulation standards. These agents also may have negative effects on animals, but according to product data sheets, this was only noticed at a concentration >5%. Moreover the 70-100% biodegradability is reached within 1 month which is a positive aspect of using such material in the environment.”
Discussion:
“Also not all wet-green tanning agents are safe for the aquatic environment, therefore a thorough examination of the final fertilizer composition from wet-greens should be done prior to soil application an decided upon the national regulation standards”.
- It is suggested to show chemical characterization about elemental or atomic composition or further experiments about nitrogen content because this section is not so clear to me. Authors could do some scanning electron microscopy experiments or any analytical technique to get information about the elements considered such as X-ray photoelectron spectroscopy or another
Answer: Thank you for this comment. Unfortunately we do not have the access to SEM or X-ray photoelectron spectroscopy. We added the Table 3S with some elemental composition of the materials used.
- English could be revised
Answer: Thank you for this comment. English was revised.
Reviewer 3 Report
Comments and Suggestions for Authors
Title:
Recovery of nitrogen from tanned leather waste in ryegrass growth after waste hydrolyzation, carbonization and microbial treatment
Recommendation:
Accept after minor revisions.
Comments:
This manuscript discusses the recovery of nitrogen from tanned leather waste and its potential use as an organic fertilizer. It explores various treatment methods, including hydrolyzation, carbonization, and microbial treatment, to enhance the growth of ryegrass. The research emphasizes the environmental benefits of recycling leather waste, reducing the reliance on inorganic fertilizers, and minimizing greenhouse gas emissions. Additionally, it addresses the challenges posed by chromium contamination in leather waste and suggests safe practices for utilizing such materials in agriculture, contributing to a circular economy. The subject is relevant and consistent with the aims and scopes of the journal. Several comments and suggestions are offered below with the intent to assist the author in improving the manuscript.
1. In this manuscript, it is recommended that Tables 1 and 2 in this manuscript be moved to the supplementary section.
2. The entire manuscript should be revised to align with the concise style typical of scientific articles, particularly in the Results section (Chapter 3).
3. In this study, how can the findings of this study be applied to other crops or agricultural practices to enhance sustainability?
4. In this study, what further research is needed to fully understand the implications of using tannery waste in agriculture, especially concerning soil health and food safety?
Author Response
REVIEWER 3
This manuscript discusses the recovery of nitrogen from tanned leather waste and its potential use as an organic fertilizer. It explores various treatment methods, including hydrolyzation, carbonization, and microbial treatment, to enhance the growth of ryegrass. The research emphasizes the environmental benefits of recycling leather waste, reducing the reliance on inorganic fertilizers, and minimizing greenhouse gas emissions. Additionally, it addresses the challenges posed by chromium contamination in leather waste and suggests safe practices for utilizing such materials in agriculture, contributing to a circular economy. The subject is relevant and consistent with the aims and scopes of the journal. Several comments and suggestions are offered below with the intent to assist the author in improving the manuscript.
- In this manuscript, it is recommended that Tables 1 and 2 in this manuscript be moved to the supplementary section.
Answer: Thank you for this comment. Both tables were moved to the Supplementary Material
- The entire manuscript should be revised to align with the concise style typical of scientific articles, particularly in the Results section (Chapter 3).
Answer: Thank you for this comment. The text was revised.
- In this study, how can the findings of this study be applied to other crops or agricultural practices to enhance sustainability?
Answer: Thank you for this comment. The following text was added to the Discussion section:
“The results of this study could also find application for other crops not designated for the direct consumption such as grasses, energy crops, flowers, trees, maize, ocra and pine. Light hydrolysates could possibly be used for the vegetables, rice etc. simply because the risk of carrying toxic compounds is very low. Heavy hydrolysates after collagen extraction due to the harmful effects on ryegrass growth in the first 3 months, should not be used for any of the cases, unless additionally pretreated. Wet-greens based materials due to potential harmful effects in aquatic environment should be thoroughly examined prior to use in the larger scale, however their use in the controlled conditions eg. in the greenhouse could find an application. This is because the negative effects on animals were only noticed at concentration >5% and the 70-100% biodegradability is reached within 1 month.”
- In this study, what further research is needed to fully understand the implications of using tannery waste in agriculture, especially concerning soil health and food safety?
Answer: Thank you for this comment. The following text was added to the Discussion section:
“Future research directions could be centred around validation of toxicology aspects when heavier waste are used, further microbially induced nutrients bioavailability stimulation, effects on growths and N uptakes over longer periods of time, speciation of N forms and distribution in soil, roots and plant fractions, applying other crop-ping systems and under different climatic conditions, optimising collagen extraction with maintaining safer extracts and byproducts which can be directed back to the soil.”
Round 2
Reviewer 1 Report
Comments and Suggestions for Authors
Can be considered.
Comments on the Quality of English LanguageThe use of the definite/indefinite article corrections is not right.
Reviewer 2 Report
Comments and Suggestions for Authors
Authors revise manuscript although there is still needed to revise typographic errors and english grammar. Manuscript can be published after correction
Comments on the Quality of English LanguageSome paragraphs need revision of english grammar